# Intrahousehold Food Intake Inequality by Family Roles and Age Groups

**DOI:** 10.3390/nu15092126

**Published:** 2023-04-28

**Authors:** Khatun Mst Asma, Koji Kotani

**Affiliations:** 1Research Institute for Future Design, Kochi University of Technology, Kochi 782-8502, Japan; asmakhatun.bau@gmail.com; 2Department of Agricultural and Applied Statistics, Bangladesh Agricultural University, Mymensingh 2202, Bangladesh; 3School of Economics and Management, Kochi University of Technology, Kochi 782-8502, Japan; 4Urban Institute, Kyusyu University, Fukuoka 812-8582, Japan; 5College of Business, Rikkyo University, Tokyo 171-8501, Japan

**Keywords:** dietary diversity, family role, age, intrahousehold inequality, Bangladesh

## Abstract

Food intake inequality at the individual level is rarely analyzed in intrahousehold settings. We examine dietary diversity scores of household members with a focus on their family roles (fathers, mothers, sons, daughters and grandparents) and age groups (children, adults and elderly). Whereas theory suggests that members in a household should have equal dietary diversity by receiving a certain share of available foods, this research hypothesizes that they do not do so by their roles and/or age groups. We conduct questionnaire surveys, collecting sociodemographic information and dietary data by using a 24 h recall method of 3248 subjects in 811 households from 1 urban and 2 rural areas in Bangladesh. The statistical analysis demonstrates three findings. First, poor and rural people have lower dietary diversity than nonpoor and urban people, respectively. Second, grandparents (children) have lower dietary diversity than do fathers (adults), confirming the existence of intrahousehold food intake inequality by the roles and/or age groups, irrespective of poverty level and areas of residence. Third, father and mother educations are crucial determinants that raise the dietary diversity of household members; however, they do not resolve the inequality. Overall, it is suggested that awareness programs of dietary diversity shall be necessary with a target group of fathers and mothers for the betterment of intrahousehold inequality and health at the household level, contributing to sustainable development goals.

## 1. Introduction

Nutritional deficiency is one of the severe problems around the globe, especially in developing countries. It is also reflected in the sustainable development goals (SDGs) that emphasize the need for special attention to eradicate the malnutrition problem. Recently, intrahousehold food allocation has become a priority for researchers, policy planners and development practitioners because household food adequacy does not imply the nutritional adequacy of individuals [1]. Individual nutritional status largely depends on the food allocation among household members [1]. Inequality in intrahousehold food intake is one of the major processes that exacerbates nutritional deficiencies in certain subgroups of the population within households [1,2]. Therefore, it is important to understand the dimensions of food intake inequality among household members and to identify the vulnerable subgroups of population at the intrahousehold level. Such an understanding will support designing appropriate policies and enhancing equitable food intake within households to improve nutritional status and contribute to SDGs. The present study addresses intrahousehold food intake inequality by subgrouping household members according to their family roles and age groups.

The literature analyzes intrahousehold adequacy in food intake with respect to sex, focusing on specific age groups [3,4,5,6,7]. Hossain et al. [8] establish sex bias in food intake and show that energy and protein consumption is higher for sons than for daughters among empowered women households in Bangladesh. Aurino [9] demonstrates that boys are more advantaged in terms of intrahousehold food allocation than girls, particularly for children and adolescents in India. Similarly, Akerele [1] shows that adult males consume more energy than others in Nigeria. Harris-Fry et al. [4] find that male household heads have higher dietary adequacy and consume more animal-sourced foods than household women in Nepal. Singh [10] assesses intrahousehold food discrimination in India and finds that sex has a significant effect on child nutrition. However, Finaret et al. [11] examine dietary patterns of children within households in Nepal and demonstrate that there are not sex biases but age biases in intrahousehold food allocation. Overall, these studies establish sex discrimination in intrahousehold food intake by focusing on certain age groups.

Another group of studies examines food intake patterns and dietary practices in relation to sociodemographic characteristics through questionnaire surveys at individual and/or household levels. Fernandez-Alvira et al. [12] assess the relationship between parental education and children food intake behaviors in Europe and show that parental education has an effect on healthy dietary practices. Rabbani [13] compares the dietary diversity of poor and nonpoor households in Bangladesh by using secondary data and concludes that dietary diversity is lower in poor families than in nonpoor families. Bose and Dey [14] examine household diversity patterns in rural and urban areas of Bangladesh and find that households suffer from food poverty not by cereals but by pulses, livestock and horticulture commodities in both areas. Ponce et al. [15] find that the urban poor have higher dietary diversity than the rural poor in Mexico. Jayawardena et al. [16] and Keino et al. [17] estimate individual dietary diversity and its relation to sociodemographic factors in Sri Lanka and Kenya, respectively, and they report that age, sex, area of residence, education and ethnicity are highly correlated with diversity. Overall, these studies suggest that sociodemographic characteristics are important determinants for explaining food intake patterns and diversity practices, regardless of the country.

Most of the prior studies have examined food intake practices and patterns based on sex and specific age cohorts by selecting a subgroup of the population at the household level. However, there are few studies to address dietary diversity inequality at the individual level in intrahousehold settings. Intrahousehold inequality can be guided by parental investment (PI) theory. According to PI theory, when parents decide to invest, they need to think about whether it is more advantageous to invest finite resources in offspring and other kin, mates or themselves [18]. PI theory hypothesizes that parents experience a fitness trade-off between quality and quantity of offspring and their care. Based on PI theory, we hypothesize that there exists an inequality of dietary diversity by their family roles (fathers, mothers, sons, daughters and grandparents) and age groups (children, adults and elderly). Given the scarcity of the literature, we analyze dietary diversity scores (DDSs) of all members per household along with sociodemographic factors in a single framework. Specifically, we seek to answer the following open research questions: (i) How do household members have dietary diversity by their roles and/or age groups, depending on poverty level and area of residence? and (ii) Who are the vulnerable food intake subgroups within households?

## 2. Methods

### 2.1. Survey Areas, Sample and Sampling Strategy

A cross-sectional design was applied to collect data from multiple members of households with a predefined questionnaire in three districts, namely, Dhaka, Jashore and Satkhira in Bangladesh, during the period between February 2019 and June 2019 (see Figure 1) The Dhaka district is an urban and densely populated area, while the Jashore and Satkhira districts are rural and less densely populated areas in Bangladesh. The current study randomly identified 900 households, specifically, 300 from Dhaka, 300 from Jashore and 300 from Satkhira. However, 874 households provided all information contained in the questionnaire, while 26 households had missing observations in urban and rural areas. We excluded the households and the associated members with such missing observations from our analysis. In total, 811 households, that is, 219 from urban and 592 from rural areas, were selected for the final analysis. Among the selected households, in total 3248 (94%) subjects participated in the surveys. The median age of the subjects is 29 years, ranging from 2 to 102 years old. The number of subjects per household ranges from 2 to 9, with a median of 4. The data of children aged between 2 and 10 years were collected from their mothers because they get involved in the food preparation and take care of their children. Sometimes, other family members, such as children or grandmothers, help mothers in preparing and serving food in Bangladesh. (Pregnant women, children aged below 2 years and people who suffers from some diseases that are related to food intakes were not considered subjects from the beginning of our surveys). Among all the children, 74% children reported their own information during the surveys.

The data collection procedures follow a hierarchical nature where subjects are nested into households. In this study, we applied two different methods for random sampling between urban and rural areas, because these areas have different geographic and sociodemographic characteristics [19]. In the urban area, we applied an occupation-based randomization technique to represent the population. Dhaka, the urban area, is the most densely populated city in the world with high mobility and a number of slums and day workers. Therefore, accumulating all correct or precise information of residents from city offices within Dhaka is not feasible due to the aforementioned nature of the city. If we sent invitation letters to residents in Dhaka for participation based on usual household-based randomization procedures, the participation rate would be very low or skewed to a certain group of people due to the fact that letters may not correctly arrive or go to the wrong places depending on the social classes of people. For instance, household-based sampling procedures do not allow us to include low-income people as subjects due to their frequent movement within the city. Consequently, we would not be able to include a wide variety of people in samples and are likely to suffer from selection biases. Therefore, we implemented randomization and sampling based on occupations to include a variety of households, i.e., even those who reside in slum areas through the channels of local NGOs or some organizations [19,20]. With this technique, first, we computed the proportion of each occupation on the basis of previous reports by governmental authorities in Bangladesh [21]. Second, we proportionally identified the required number of households from randomly selected organizations, such as different government and nongovernmental organizations, based on each occupation.

In rural areas, the list of households who reside in Jashore and Satkhira districts was collected in cooperation with local nongovernmental organizations (NGOs). By using this list and a random number generator, we selected the required number of households from each rural area. Trained research staff contacted the selected households and obtained sociodemographic and dietary intake information by conducting our survey questionnaires. The research staff were carefully trained about how to conduct the surveys. Regarding the informed consent, we took the following steps. At first, the household head (a husband or wife in a household) took the verbal consent of all household members before starting our surveys. If all the household members agreed to participate in our surveys, we collected written consent from only the head of the household as a guardian of the household. The trained research staff followed a face-to-face data collection method and the total questionnaire session took almost 1 h to complete. There was a fixed participation fee: 50 BDT per household. Individual information was collected from each subject separately. In case of dietary information, each subject reported their self food groups consumption except subjects aged between 2 and 10 years. The first author served as the chief administrator and monitored the surveys.

### 2.2. Key Variables

Dietary diversity is concerned with the number of food groups consumed by a person in a given period of time [9,22,23]. To measure the dietary diversity score (DDS) per subject, data on food items are categorized into 9 food groups by following the food and agriculture organization (FAO) guidelines [24] as follows: (i) starchy staples; (ii) dark green leafy vegetables; (iii) other vitamin A rich fruits and vegetables; (iv) other fruits and vegetables; (v) organ meat; (vi) meat and fish; (vii) eggs; (viii) legumes, nuts and seed; and (ix) milk and milk products. A dummy variable is created for each food group and assigned the values of 0 and 1. If a subject consumes any item from a particular food group, the subject is assigned a value of 1; otherwise, the subject is assigned 0. A set of 9 food groups is used to calculate the DDS by adding the number of food groups consumed by each subject in a period of the past 24 h. The maximum value of DDS is 9 and the minimum value is 0. To calculate the DDS, we assign a 15 g minimum quantity or one tablespoon, which is a subjective measure [24,25]. For example, if subjects think that the quantity of any food item consumption is less than 15 g or one tablespoon, we do not include them in the DDS. The diversity calculation with the 9 food groups adopted by the FAO is established to perform well in developing countries to reflect micronutrient adequacy as well as overall intakes at the individual level [26,27]. Therefore, in this study, the DDS is used as a measurement of dietary diversity per subject.

Each household member’s nuclear role within a household was asked to be confirmed and the data were recorded. Chan and Sobal [28] apply the same procedure to identify the role of each household member within a household. In the present study, five types of family roles such as fathers, mothers, sons, daughters and grandparents were identified based on each household member’s self-reported specific role as well as another reconfirmation from other members in the household. A husband or a wife is a household head and their family roles are categorized as fathers or mothers, respectively. Young and adult children usually have the roles of sons and daughters within households. Household members who reported their family role as grandparents are basically grandfathers and grandmothers. We separate grandparents into grandfathers and grandmothers at the end of our study to identify any gender differences in dietary diversity at grandparents’ level.

The present study estimates poverty in both urban and rural areas based on the cost of basic need (CBN) method [29]. The CBN method represents the level of per capita expenditure of a household to meet the basic needs of its members including both food and nonfood allowances. The nonfood allowance includes expenditures of fuel and lightning, transport and travel, clothing, health, housing, education, recreation and leisure [29]. At first, the expenditure of a food bundle is computed, which includes rice, wheat, pulses, milk, oil, meat, fish, potato, other vegetables, sugar, fruits and others. Next, nonfood expenditures of household are calculated and added to the food expenditure to obtain the total expenditure of a household. To total expenditure is then divided by the household size to calculate per capita expenditure per month. Specifically, with this method, the poverty line indicates the minimum average level of per capita expenditure below which a household cannot meet their basic food and nonfood needs. The CBN approach is known as an official methodology for estimating poverty in Bangladesh, where a household under absolute poverty is one whose per capita expenditure is below the upper poverty line (See Appendix A). The estimated upper poverty lines are 2929 BDT (36.08 USD) and 2019 BDT (24.87 USD) in the selected urban and rural areas, respectively [29]. In this study, a household is defined as poor if its per capita monthly expenditure (nonfood and food) is less than the national estimated upper poverty line; otherwise, it is defined as nonpoor.

During the questionnaire surveys, information was collected on age, area of residence, father education, mother education, total household earners, occupation of the household head, religion, family structure and household eating practices. Some of the literature finds that the relationship between age and the DDS is not linear [11,30]. Therefore, the age of the subjects is categorized into three groups: children (below 16 years old), adults (between 16 and 60 years old) and elderly individuals (above 60 years old), and we create separate dummy variables to accommodate the possible nonlinearity in the analysis, following the past literature [31,32,33]. According to different laws, children age varies between 14 and 18 years. The children act of Bangladesh defines a person below 16 years of age as a child [32]. In this study, approximately 50% of the data overlap between grandparents and the elderly. To this end, we conduct questionnaire surveys, collecting sociodemographic information and dietary data by using a 24 h recall method of 3248 subjects in 811 households from one urban and two rural areas in Bangladesh. Table 1 represents the descriptions of all variables used in this study.

### 2.3. Statistical Analysis

We compute and interpret the descriptive statistics, such as mean, median and standard deviation of the dependent and independent variables. We use chi-squared test to identify the associations between food groups consumption and family roles. The F test is applied to examine whether or not the means of the DDS differ by family roles. We also apply chi-squared and Mann–Whitney tests to compare the differences in the key variables by urban and rural areas. The following pairs are considered: (i) children’s DDS vs adults’ DDS (ii) elderly’s DDS vs adults’ DDS by applying the Mann–Whitney test. A Wilcoxon matched-pairs singed-rank test is implemented to assess the paired differences of the DDS between fathers and other household members (mothers, sons, daughters and grandparents). The following pairs are tested: (i) fathers’ DDS vs. mothers’ DDS (ii) fathers’ DDS vs. sons’ DDS (iii) fathers’ DDS vs. daughters’ DDS and (iv) fathers’ DDS vs. grandparents’ DDS.

To quantitatively identify the inequality of the DDS among household members based on their family roles and age groups, we employ a Poisson regression in our analysis due to the positive and count values of the DDS [34,35]. The ordinary Poisson regression can be specified as follows:(1)$$\ln\left(\mu_{i}\right)=\beta_{0}+\alpha F_{i}+\beta A_{i}+\gamma X_{i}+\varepsilon_{i}$$
where $\mu_{i}$ is the expected value of DDS for *i*th subject, $F_{i}$, $A_{i}$ and $X_{i}$ are the vectors of the family role dummies, age group dummies and sociodemographic variables, respectively, and $\varepsilon_{i}$ is an error term. The $\beta_{0}$ is the parameter associated with the intercept, while $\alpha =(\alpha_{1},\alpha_{2},\ldots ,\alpha_{5})$, $\beta =(\beta_{1},\beta_{2})$ and $\gamma =(\gamma_{1},\gamma_{2},\ldots ,\gamma_{9})$ are the vectors of the parameters associated with $F_{i}$, $A_{i}$ and $X_{i}$, respectively. In this research, we are interested in estimating the coefficients of α and β in Equation (Equation 1). We can interpret the coefficients of explanatory variables in the Poisson regression in the following way. Suppose an estimated coefficient of each sociodemographic variable, ${\hat{\gamma }}_{j},j=1,2,\ldots ,9$ is considered to represent the marginal effect of this variable on the DDS after the effects of the other variables are netted out. The marginal effect of a continuous explanatory variable, such as father education, is derived from a formula ${\hat{\gamma }}_{1}\times 100$ to be a percentage change in the expected value of the DDS with a one-year increase in father education. A dummy explanatory variable, such as household poverty (poor=1 and nonpoor=0), is calculated by $[\exp\left({\hat{\gamma }}_{3}-1\right)]\times 100$. It is interpreted as a percentage change of the expected value of the DDS when the household poverty increases from 0 to 1 (see, e.g., [35]).

The subjects are nested (or clustered) by households, and thus, the ordinary Poisson regression model is customized to consider the cluster-specific effect in the model. The simplest modification is called the two-level random intercept Poisson regression model in which the intercept captures the cluster-specific effect from the other covariates [36]. The multilevel model provides efficient estimates and captures the unobserved variation in the model [37,38]. Moreover, multilevel modeling is employed to differentiate the individual and household level characteristics for the relationship between independent and dependent variables [28]. The two-level random intercept Poisson regression model that consider subjects at level 1 and households at level 2 can be written as follows:(2)$$\ln\left(\mu_{ik}\right)=\beta_{0}+\alpha F_{ik}+\beta A_{ik}+\gamma X_{ik}+\varepsilon_{0k}+\varepsilon_{ik}$$
where $\mu_{ik}$ is the expected value of DDS for *i*th subject living in *k*th cluster (household). $F_{ik}$, $A_{ik}$ and $X_{ik}$ are the vectors of the family role dummies, age group dummies and sociodemographic variables, respectively, for the *i*th subject in the *k*th cluster (household). The regression coefficient $\beta_{0}$ is the intercept, while the coefficients $\alpha =(\alpha_{1},\alpha_{2},\ldots ,\alpha_{5})$, $\beta =(\beta_{1},\beta_{2})$ and $\gamma =(\gamma_{1},\gamma_{2},\ldots ,\gamma_{9})$ are the vectors of the parameters associated with $F_{ik}$, $A_{ik}$ and $X_{ik}$, respectively. The $\varepsilon_{0k}$ is a cluster-specific random component that is assumed to be independently and normally distributed, and $\varepsilon_{ik}$ is an error term. The interpretation of the regression coefficients in a two-level random intercept Poisson regression remains the same as an ordinary Poisson regression model, but the intercept interpretation is different [39]. The cluster-specific random component can capture the unobserved variation in the model that is not explained by the explanatory variables. If the cluster-specific effect is significant in the model, then we conclude that subjects from different households with the same set of values and levels of the independent variables will show different DDSs.

A series of ordinary Poisson and the two-level random intercept Poisson regression models are applied step by step to check the robustness of the results. First, we include the family role dummies with fathers as the base group in Model 1-1 and Model 2-1 in the ordinary Poisson regression and the two-level random intercept Poisson regression, respectively. Then, we exclude the family role dummies and include the age group dummies with adults as the base group in Model 1-2 and Model 2-2 in the ordinary Poisson regression and the two-level random intercept Poisson regression, respectively. Finally, we include all the independent variables, such as family role dummies, age group dummies and sociodemographic variables in Model 1-3 and Model 2-3 in the ordinary Poisson regression and the two-level random intercept Poisson regression, respectively.

## 3. Results

Table 2 summarizes the descriptive statistics, such as mean, median and standard deviation of the DDS and food group consumption by the family roles. The mean DDS for the overall sample is 4.88 (see the “overall” column in Table 2). Food groups such as starchy staples; dark green leafy vegetables; other vitamin A rich fruits and vegetables; and meat and fish are mostly consumed, while the consumption of animal sources of foods, such as organ meat; eggs; milk and milk products; other fruits and vegetables and legumes, nuts and seeds are consumed in relatively low quantities by subjects, irrespective of the family roles. The analysis reveals that consumption of food groups among household members are correlated with their family roles except starchy staples, other vitamin A rich fruits and vegetables and organ meat. From Table 2, the results confirm that households members have different DDSs with their family roles.

Table 3 presents the summary statistics, such as mean, median and standard deviation, of the key dependent and independent variables for urban and rural areas. A major difference is observed in the mean DDS in urban and rural areas. Urban subjects (5.61) have significantly higher dietary diversity than rural subjects (4.63). Considering the family roles, 26% of the urban population is mothers, while this percentages is 24% in rural areas. In case of the base group of family roles, the percentages of father are 26% and 24% in the urban and rural areas, respectively. However, significant differences exist in the percentages of grandparents between the urban (3%) and rural areas (6%). Based on the age groups, the percentages of children and elderly (22% and 5%) are higher in rural areas than the urban area (17% and 3%). There are variations in father and mother educations between the urban and rural areas. The median of father education in the urban area is 12 years of schooling, while this median is 7 years of schooling in rural areas. The median of mother education in the urban area is 11 years of schooling, while this median is 7 years of schooling in rural areas. Regarding household poverty, 17% of the overall subjects are considered poor people and the largest variation is found in the percentages of poor people who live in the urban and rural areas. In the urban area, 5% of the subjects live below the poverty line, while this percentage is 22% in rural areas.

The median number of total household earners is one in both the urban and rural areas in Table 3. Regarding occupation, all urban household heads are engaged nonagricultural activities, while 41% of the rural household heads are engaged in agricultural activities. Table 3 also shows that 90% of the urban subjects are Muslim, while 87% of the subjects are Muslim in rural areas. The main family structure of the overall sample in both areas (urban and rural) is the nuclear family; however, the percentage of the extended family is relatively higher (30%) in rural areas than the urban area (19%). In terms of household eating practices, 68% of urban subjects and 85% rural subjects have a practice to eat together with their family members. In Table 3, some of the variables, such as total household earners and religion, are statistically significant even though their mean difference is very small. A possible reason behind this is the small variability in the samples. Based on our estimation, total household earners and Muslim people are relatively higher in the urban area than rural areas. In summary, rural areas have lower dietary diversity but a higher number of grandparents, children and elderly individuals than the urban area. Most of the sociodemographic variables, such as poverty, education, earners, occupation, religion, family structure and eating practices, vary between urban and rural areas.

Figure 2a shows the boxplots of DDSs of household members by their family roles, and Figure 2b presents the boxplots of DDSs by their age groups. In Figure 2a, the DDS distribution in grandparents is lower than that in fathers with respect to the medians. The boxplots cannot capture the differences between fathers’ DDS with other household members’ DDS. Therefore, we apply a Wilcoxon matched-pairs singed-rank test to compare the distributional differences of fathers’ DDS with other household members’ DDS. The null hypothesis is that the distributions of DDS between fathers and mothers pairs are the same. The test results show that the distributions of DDS between fathers and mothers (Z=3.36, p=0.01), fathers and sons (Z=2.07, p<0.04), fathers and daughters (Z=1.64, p<0.10) and fathers and grandparents (Z=4.96, p<0.01) are not the same. In Figure 2b, it can be seen that the DDS distributions in children and elderly individuals are lower than those in adults with respect to the medians. We run a Mann–Whitney test with a null hypothesis, which shows that the DDS distributions between children and adults are the same. We find that there is a distributional difference in the DDS between children and adults (Z=−4.61, p=0.01) but we do not find any distributional difference in the DDS between the elderly and adults (Z=1.59, p=0.11).

The descriptive statistics, tests and diagrams suggest that the DDS varies among household members by their family roles and/or age groups. We run ordinary and two-level random intercept Poisson regressions to further characterize the relationships of the DDS to the family role and age group dummies after controlling for sociodemographic variables. Table 4 reports the estimated coefficients of the explanatory variables on the DDS in the ordinary and two-level random intercept Poisson regressions with several model specifications.

Sociodemographic variables, such as father education, mother education, household poverty, area and total household earners, are identified to be statistically and economically significant in Model 1-3 and Model 2-3 in the ordinary Poisson regression and the two-level random intercept Poisson regression, respectively (see Table 4). The effects of father and mother educations are generally demonstrated to have a positive effect on their household nutrition. In terms of father education, the ordinary Poisson regression and the two-level random intercept Poisson regression in Model 1-3 and Model 2-3 find that the expected DDS increases by 0.7% and 0.7% per one-category increase in schooling. In the case of mother education, the expected DDS increases by 1% and 1% per one-category increase in schooling (see Table 4). Overall, we corroborate that there is a positive relationship between education and healthy food practices in intrahousehold settings.

The regression results of household poverty in Model 1-3 and Model 2-3 find that the expected DDS of the poor is 15% and 15% lower than that of the nonpoor, holding other factors fixed. Likewise, the results of the area dummy in Model 1-3 and Model 2-3 can be interpreted. The expected DDS of rural subjects is calculated to be 12% and 12% lower than that of urban subjects. (Household poverty and area are dummy variables, therefore, we use the following formula to calculate the marginal effect of household poverty and area, respectively: exp(0.14)−1≈0.15=15% and exp(0.11)−1≈0.12=12%). The number of total household earners has an effect on the DDS, i.e., the ordinary Poisson regression and the two-level random intercept Poisson regression in Model 1-3 and Model 2-3, respectively; the models estimate 5% and 5% increases in the expected DDS per an increase in earners within the household.

Models 1-3 and 2-3 examine the effects of the family role and age group dummies on the DDS in Table 4: more precisely, they control for other sociodemographic variables. The regression results of the family role and age group dummies in Model 1-3 and Model 2-3 do not differ from those of Model 1-1 and Model 2-1, confirming the consistency and robustness of our results. The family role and age group dummies are identified as important determinants of the DDS in both the ordinary and two-level random-intercept Poisson regressions, with 5% statistical and economic significance. The ordinary Poisson regression and the two-level random intercept Poisson regression estimation in Model 1-3 and Model 2-3 reveal that the expected DDSs of grandfathers and grandmothers are 19%, 14% and 16%, 13% lower than those of fathers. Likewise, the results of the age group dummies in Model 1-3 and Model 2-3 can be interpreted. The expected DDS of children is calculated to be 8% and 6% lower than that of adults. (The marginal effects of the family role dummies and age group dummies are calculated by using the formula: $[\exp\left({\hat{\alpha }}_{j}-1\right)]\times 100$, where ${\hat{\alpha }}_{j}$ is the estimated regression coefficient of the dummy variable. For example, exp(0.17)−1≈0.19=19%). Consistent with the summary statistics, both regression estimations confirm the inequality in food intake among household members and identify that grandparents and children are the vulnerable food intake subgroups within households.

In Table 4, we notice that father and mother educations are important to raise the dietary diversity of household members; however, we do not find any interaction effect between family roles and either father or mother educations. After controlling for poverty, areas of residence and between household variabilities, we find that the DDS varies among household members according to their family roles and age groups. This implies that the dietary diversity inequality exists at the individual level in intrahousehold settings. Again, the cluster-specific effect is observed and significant in Models 2-1, 2-2 and 2-3 in the two-level random intercept Poisson regressions, meaning that subjects from different households with the same set of values and levels of the independent variables will show different DDSs (see Table 4). The magnitude of the cluster-specific effect is greater than the effects of some of the important explanatory variables in the models. For instance, the standard deviation of the random cluster-specific effect in Model 2-3 is 0.16, indicating that a one-standard deviation change in the cluster-specific effect has a greater effect on the DDS than household poverty ($\gamma_{3}=-0.14$). In such a situation, household characteristics, such as dietary patterns, frequencies of the main meals, nutritionally balanced foods and household members’ nutritional awareness, could be given priority to improve household dietary diversity practices.

## 4. Discussion

Now, it is time to provide answers to the following open research questions: (i) How do household members have different dietary diversity by their roles and/or age groups, depending on poverty level and area of residence? The summary statistics, tests and diagram suggest that household members do not have equal dietary diversity by their family roles and/or age groups. The regression results further show that household members have different DDSs after controlling for sociodemographic variables. Overall, it can be concluded that household members have different dietary diversity, confirming the existence of intrahousehold food intake inequality, irrespective of poverty level and area of residence. The other research question is as follows: (ii) Who are the vulnerable food intake subgroups within households? The regression results consistently show that grandparents (children) have lower dietary diversity than those of fathers (adults). This indicates that grandparents and children are the vulnerable subgroups in the case of nutritional adequacy within households.

Inequality in intrahousehold food intake is a major concern that promotes nutrient deficiencies and perpetuates the malnutrition problem [1,2]. The literature argues that family roles influence how people perceive and behave toward food and nutritional outcomes in a certain way [28,30,40]. However, it is currently unknown whether intrahousehold food intake inequality exists by the family roles and/or age groups. This research confirms intrahousehold food intake inequality and finds that grandparents and children are the vulnerable food intake subgroups within households. It is widely demonstrated that in households with some level of food insecurity, women and older people are at high risk for limited access to food; children are also the last sufferers in terms of access to food [41,42]. Some readers may wonder if the results are considered inconsistent with ours. However, recall that our focus in the analysis is on the diversity of food, not on the quantity, meaning that our results do not necessarily contradict the existing findings. Now, it is time to answer the question: “why do household members have significantly different food intakes?” We can explain intrahousehold food intake inequality in the following three ways: (i) Parental investment (PI) theory, (ii) Contribution rule and (iii) and the lack of mothers’ nutritional knowledge.

Intrahousehold inequality can be explained by principles of PI theory. PI theory focuses the evolution of age and sex differences, resulting in a taxonomic bias in the pattern of parental care [18,43,44]. Parents need to think about whether it is more advantageous to invest finite resources in offspring and other kin, mates or themselves when they decide to invest [18]. PI theory hypothesizes that parents experience a fitness trade-off between quality and quantity of offspring and their care. For example, parents allocate more investment to their older children than infants in case of long-term stress; however, they show the opposite behavior in case of short-term stress [18]. Blum et al. [45] indicate that adult males and boys are more favored for food and nutrition than others because they are expected to provide money and care for their parents currently and in their old age. Based on PI theory, we can explain the unequal dietary diversity that parents (mainly, mothers) prefer to older children over younger ones in food allocation, creating intrahousehold food intake inequality.

Intrahousehold food intake inequality can be explained by using the “contribution rule”. This describes a situation where household members who contribute to the family are more likely to be favored for food and nutrition than other family members [3]. For example, Harris-Fry et al. [4] document that families invest more nutritious food in economically productive members, which results in higher incomes. According to the “contribution rule”, it is considered that household heads and adults are favored in terms of food and nutrition, because they are the main sources of financial support and security to the household. On the contrary, grandparents and children are not favored because they are considered to have no contribution to the household. We argue that the “contribution rule” implicitly remains part of food cultures in Bangladesh and is applicable to explain the existence of intrahousehold food intake inequality. At the same time, we note that household members should have equal dietary diversity (not the amount of food intake) and receive a certain share of available foods in a well-balanced manner for the betterment of health in theory [46].

It is well documented that in developing countries, a household woman, i.e., a mother, is the sole decision maker in the preparation, serving and allocation of food. They are also responsible for the care of household members [42]. In reality, it is very difficult to measure the relative size of dietary needs as well as nutritional requirements for every household member, especially for children and older people. Moreover, there are many misconceptions regarding nutritional knowledge. For example, one statement regarding nutrition is that children consume nearly one-fourth as much food as they require in adult life [11]. This type of perception may create difficulty in ensuring equitable diversity and food intake within households. Food allocation among household members is also related to mothers’ knowledge, attitudes and practice [47]. Several studies document that mothers’ nutritional knowledge is positively related to receiving a good nutritious diet from family members [48,49,50]. If this is the case, then we conjecture that a lack of mothers’ nutritional knowledge shall be responsible for intrahousehold food intake inequality. Therefore, it is recommended to specifically examine how mothers’ nutritional awareness is related to intrahousehold food intake to take good nutritional care of household members.

Poverty and areas of residence have direct effects on the dietary diversity. Irrespective of the poverty status, households in Bangladesh have an inadequate amount of food diversity. Households with a lack of adequate income or with lower per capita income are at greater risk of food insecurity [42]. It is true that educational or promotional strategies for improving dietary diversity are less likely to achieve success in the context of inadequate food affordability. Recommendations should be carefully crafted in the population with a low income. In our study, only 17% of the overall subjects are considered poor people. However, after controlling for poverty status, the inequality of food diversity remains within households by their family roles and age groups. The existence of intrahousehold food inequality might get attributed to own food choice and food availability. It is reported that food availability within households is dependent upon a number of interrelated factors, such as budget, storage facilities and household members’ food choices; however, availability of food does not necessarily lead to the improvement of dietary diversity [51]. At the same time, food choices are largely determined by taste that is opposed to any consideration of nutrition or food safety. An understanding the effects of food availability and food choices on dietary diversity can help to explain household members eating patterns as well as identify the inconsistency with their dietary recommendations. However, our data cannot support the identification of any effect on inequality in food intake due to food access and availability within households. Again, the cultural perceptions of food consumption may affect diversity patterns between urban and rural households. For example, a meal consisting of rice and eggs may be sufficient in rural areas but these food items are considered side dishes in urban areas [42]. These findings indicate the importance of empowering households both economically and socially for the betterment of nutrition and health.

The study has several implications in the field of research and policy formulation, regardless of a focus on developed and developing countries. Countries’ health and nutrition policies always focus on the underprivileged population to improve their nutritional status. In this study, daughters have lower food diversity than fathers; however, this difference does not exist after controlling for other sociodemographic variables in the model. This means that the dietary diversity of fathers and daughters does not vary significantly between them. Although females are reported to be disadvantaged in food allocation [4,8,40,52], this study does not find any sex difference in dietary diversity. The reason may be that the government of Bangladesh takes many actions regarding sex difference and inequality. We confirm that grandparents and children are vulnerable groups in terms of food diversity within households; therefore, it is necessary to focus on improving their dietary diversity for reduction of intrahousehold food inequality. However, policies should be undertaken at the government level to empower households both economically and socially for improving their nutrition and health status. We also find that father and mother educations have effects to improve the dietary diversity of household members. Therefore, systematically organizing awareness and education programs of diversity practices at household level shall be necessary for the resolution of intrahousehold food intake inequality with a target of fathers and mothers for the betterment of nutritional and health status and the contribution to SDGs.

We list some limitations of our study and provide some guidelines for future research. First, we use a 24 h recall method to calculate DDSs; however, multiple dietary recalls including both weekdays and weekends may be considered an alternative way to obtain a good picture of the habitual food intake for household members. Second, applying a 24 h recall method may suffer from reporting and recall biases. However, several studies mention that the DDS uses a 24 h recall method is reliable enough to measure individual nutrient adequacy without being significantly biased [24,53]. Third, there may be additional determinants of DDSs, such as nutritional knowledge, awareness, food preferences, availability, choices and health-related variables, that are not included in this study. We could not collect these data due to several constraints that we faced with respect to time, subjects and budgets. More detailed data about multiple dietary recalls, nutrition, health and disease-related characteristics should be considered in future studies, which would enable us to have panel data to fully characterize intrahousehold food intake inequality. These caveats notwithstanding, it is our belief that the findings of our study are sufficiently robust and are the first important step that includes all household members and quantitatively identifies intrahousehold food intake inequality by their family roles and/or age groups.

## 5. Conclusions

The study has the following three major findings: (i) Poor and rural people have lower dietary diversity than nonpoor and urban people, respectively. (ii) Grandparents (children) have lower dietary diversity than do fathers (adults), confirming the existence of intrahousehold food intake inequality by the roles and/or age groups, irrespective of poverty level and area of residence. (iii) Father and mother educations are crucial determinants that raise the dietary diversity of household members. Different policies should be undertaken at the government level to empower households both economically and socially for improving their nutrition and health status. Overall, we suggest that specific awareness and education programs of dietary diversity shall be necessary for resolving the inequality with a target group of fathers and mothers for the betterment of nutrition and health at household level, contributing to SDGs.

## Figures and Tables

**Figure 1 nutrients-15-02126-f001:**
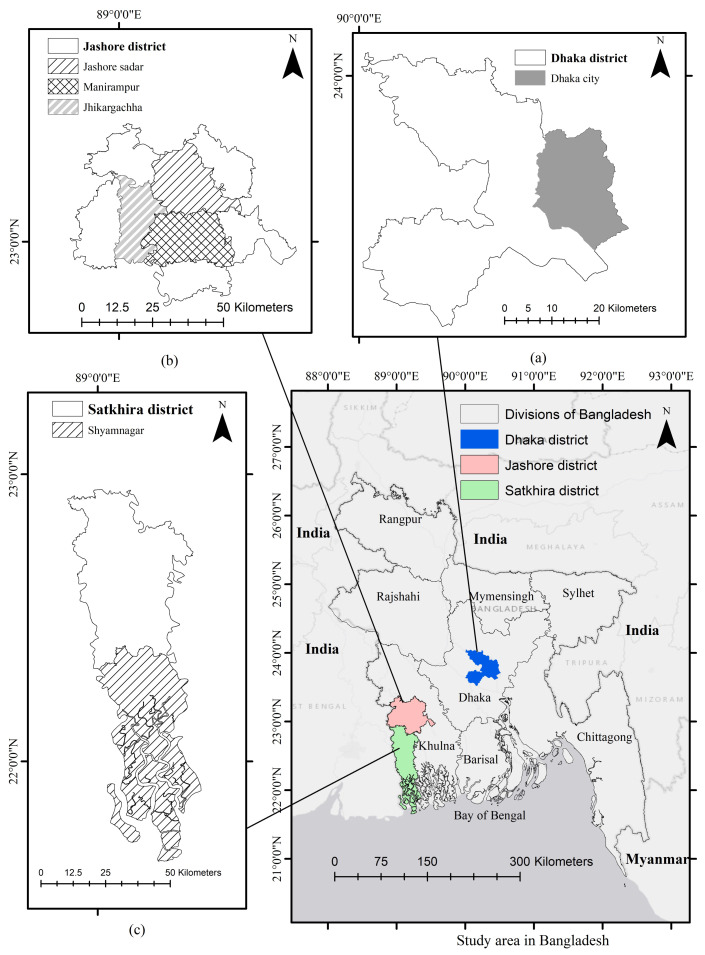
Location of study areas (districts) in Bangladesh: (**a**) Dhaka (urban), (**b**) Jashore (rural) and (**c**) Satkhira (rural).

**Figure 2 nutrients-15-02126-f002:**
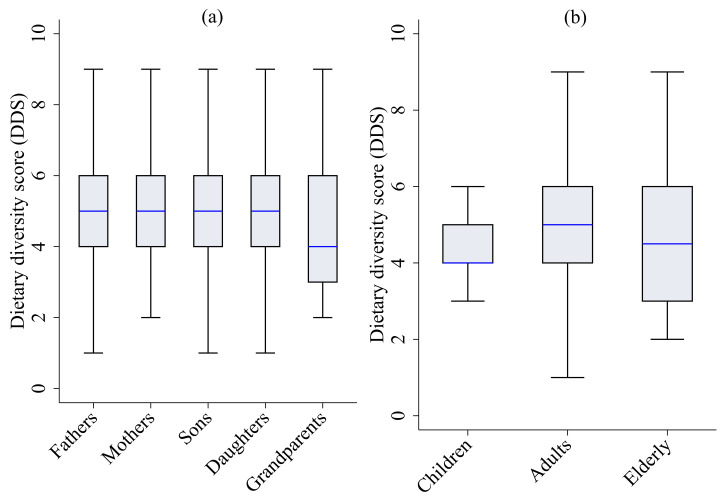
Boxplots of dietary diversity scores (DDSs) by (**a**) family roles and (**b**) age groups.

**Table 1 nutrients-15-02126-t001:** Definitions of the variables.

Variables	Description
Dependent variable	
Dietary diversity score (DDS)	The dietary diversity score is defined as a count variable that takes values from 0 to 9 based on the number of food groups consumed over a 24-h period.
Independent variables	
Family role dummy variables (Basegroup=Fathers)	
Mothers	Fathers 0 and Mothers 1.
Sons	Fathers 0 and Sons 1.
Daughters	Fathers 0 and Daughters 1.
Grandparents	Fathers 0 and Grandparents 1.
Age group dummy variables (Basegroup=Adults)	
Children	Adults 0 and Children 1.
Elderly	Adults 0 and Elderly 1.
Sociodemographic variables	
Father education	Years of schooling 0 to 14 (0 = No schooling and refused group ${}^{1}$, 1 = Class one, 2 = Class two, 3 = Class three, 4 = Class four, 5 = Class five, 6 = Class six, 7 = Class seven, 8 = Class eight, 9 = Class nine, 10 = SSC/equivalent, 11 = Eleven class/equivalent, 12 = HSC/equivalent, 13 = Graduate/equivalent, 14 = Post graduate/equivalent).
Mother education	Years of schooling 0 to 14 (0 = No schooling and refused group ${}^{1}$, 1 = Class one, 2 = Class two, 3 = Class three, 4 = Class four, 5 = Class five, 6 = Class six, 7 = Class seven, 8 = Class eight, 9 = Class nine, 10 = SSC/equivalent, 11 = Eleven class/equivalent, 12 = HSC/equivalent, 13 = Graduate/equivalent, 14 = Post graduate/equivalent).
Household poverty	Nonpoor 0 and Poor 1.
Area	Urban 0 and Rural 1.
Total household earners	Numbers.
Occupation of the household head	Nonagriculture 0 and Agriculture 1.
Religion	Nonmuslim 0 and Muslim 1.
Family structure	Nuclear family 0 and Extended family 1.
Household eating practices	Takes the value of 1 when household members eat together, otherwise 0.

${}^{1}$ The subjects who do not provide their educational qualification are the refused group. We merge the refused group with the no schooling group because most of the uneducated people refused to provide their educational level.

**Table 2 nutrients-15-02126-t002:** Summary statistics of the DDS and food groups consumption by family roles.

	Family Roles					Overall	*p*-Value
	Fathers	Mothers	Sons	Daughters	Grandparents		
Dietary diversity score (DDS)							
Average (Median) ${}^{1}$	5.00 (5.00)	4.92 (5.00)	4.93 (5.00)	4.74 (5.00)	4.44 (4.00)	4.88 (5.00)	
SD ${}^{2}$	1.62	1.60	1.54	1.50	1.48	1.57	0.01 ${}^{3}$
Starchy staples							
Average (Median)	0.99 (1.00)	0.99 (1.00)	0.99 (1.00)	0.99 (1.00)	0.98 (1.00)	0.99 (1.00)	
Frequency (SD)	795 (0.06)	801 (0.04)	798 (0.04)	683 (0.07)	161 (0.11)	3238 (0.06)	0.15 ${}^{4}$
Dark green leafy vegetables							
Average (Median)	0.72 (1.00)	0.71 (1.00)	0.69 (1.00)	0.65 (1.00)	0.68 (1.00)	0.69 (1.00)	
Frequency (SD)	572 (0.45)	572 (0.45)	550 (0.46)	445 (0.48)	111 (0.47)	2250 (0.46)	0.04 ${}^{4}$
Other vitamin A rich fruits and vegetables							
Average (Median)	0.78 (1.00)	0.78 (1.00)	0.77 (1.00)	0.73 (1.00)	0.77 (1.00)	0.77 (1.00)	
Frequency (SD)	621 (0.42)	628 (0.41)	615 (0.42)	503 (0.44)	125 (0.42)	2492 (0.42)	0.19 ${}^{4}$
Other fruits and vegetables							
Average (Median)	0.30 (0.00)	0.28 (0.00)	0.30 (0.00)	0.29 (0.00)	0.18 (0.00)	0.29 (0.00)	
Frequency (SD)	241 (0.46)	227 (0.45)	243 (0.46)	196 (0.45)	29 (0.38)	936 (0.45)	0.02 ${}^{4}$
Organ meat							
Average (Median)	0.35 (0.00)	0.35 (0.00)	0.34 (0.00)	0.31 (0.00)	0.30 (0.00)	0.34 (0.00)	
Frequency (SD)	278 (0.48)	279 (0.48)	271 (0.47)	215 (0.46)	49 (0.46)	1092 (0.47)	0.47 ${}^{4}$
Meat and fish							
Average (Median)	0.71 (1.00)	0.70 (1.00)	0.71 (1.00)	0.65 (1.00)	0.71 (1.00)	0.69 (1.00)	
Frequency (SD)	567 (0.45)	558 (0.46)	569 (0.45)	447 (0.48)	116 (0.45)	2257 (0.46)	0.08 ${}^{4}$
Eggs							
Average (Median)	0.39 (0.00)	0.37 (0.00)	0.36 (0.00)	0.41 (0.00)	0.29 (0.00)	0.38 (0.00)	
Frequency (SD)	315 (0.49)	298 (0.48)	284 (0.48)	280 (0.49)	48 (0.46)	1225 (0.48)	0.03 ${}^{4}$
Legumes, nuts and seeds							
Average (Median)	0.52 (1.00)	0.51 (1.00)	0.52 (1.00)	0.47 (0.00)	0.42 (0.00)	0.50 (1.00)	
Frequency (SD)	415 (0.50)	409 (0.50)	411 (0.50)	325 (0.50)	68 (0.49)	1628 (0.50)	0.07 ${}^{4}$
Milk and milk products							
Average (Median)	0.23 (0.00)	0.22 (0.00)	0.25 (0.00)	0.23 (0.00)	0.10 (0.00)	0.23 (0.00)	
Frequency (SD)	187 (0.42)	175 (0.41)	199 (0.43)	158 (0.42)	17 (0.31)	736 (0.42)	0.01 ${}^{4}$
Sample size	798	802	799	686	163	3248	

${}^{1}$ The median is in parentheses. ${}^{2}$ SD stands for the standard deviation. ${}^{3}$ F test is applied to examine whether or not the means of the DDS differ by family roles. ${}^{4}$ Chi squared test is applied to examine whether or not the frequencies of the food groups are independent of family roles.

**Table 3 nutrients-15-02126-t003:** Summary statistics of the dependent and independent variables for urban and rural areas.

	Area		Overall	*p*-Value
	Urban	Rural		
Dietary diversity score				
Average (Median) ${}^{1}$	5.61 (6.00)	4.63 (4.00)	4.88 (5.00)	
SD ${}^{2}$	1.78	1.40	1.57	0.01 ${}^{3}$
Family role dummies (Basegroup=Fathers)				
Mothers				
Average (Median)	0.26 (0.00)	0.24 (0.00)	0.25 (0.00)	
Frequency (SD)	217 (0.44)	585 (0.43)	802 (0.43)	0.27 ${}^{4}$
Sons				
Average (Median)	0.26 (0.00)	0.24 (0.00)	0.25 (0.00)	
Frequency (SD)	212 (0.44)	587 (0.43)	799 (0.43)	0.48 ${}^{4}$
Daughters				
Average (Median)	0.19 (0.00)	0.22 (0.00)	0.21 (0.00)	
Frequency (SD)	162 (0.40)	524 (0.41)	686 (0.41)	0.18 ${}^{4}$
Grandparents				
Average (Median)	0.03 (0.00)	0.06 (0.00)	0.05 (0.00)	
Frequency (SD)	24 (0.17)	139 (0.23)	163 (0.22)	0.01 ${}^{4}$
Age group dummies (Basegroup=Adults)				
Children				
Average (Median)	0.17 (0.00)	0.22 (0.00)	0.21 (0.00)	
Frequency (SD)	144 (0.38)	526 (0.41)	670 (0.41)	0.01 ${}^{4}$
Elderly				
Average (Median)	0.03 (0.00)	0.05 (0.00)	0.04 (0.00)	
Frequency (SD)	23 (0.16)	111 (0.21)	134 (0.20)	0.02 ${}^{4}$
Father education				
Average (Median)	10.55 (12.00)	6.04 (7.00)	7.19 (8.00)	
SD	3.96	4.47	4.77	0.01 ${}^{4}$
Mother education				
Average (Median)	9.60 (11.00)	5.99 (7.00)	6.91 (8.00)	
SD	4.11	4.01	4.33	0.01 ${}^{4}$
Household poverty (Basegroup=Nonpoor)				
Average (Median)	0.05 (0.00)	0.22 (0.00)	0.17 (0.00)	
Frequency (SD)	44 (0.22)	520 (0.41)	564 (0.38)	0.01 ${}^{4}$
Total household earners				
Average (Median)	1.51 (1.00)	1.40 (1.00)	1.42 (1.00)	
SD	0.66	0.63	0.64	0.01 ${}^{3}$
Occupation of the household head (Basegroup=Nonagriculture)				
Average (Median)	0.00 (0.00)	0.41 (0.00)	0.30 (0.00)	
Frequency (SD)	0.00 (0.00)	980 (0.49)	980 (0.46)	0.01 ${}^{4}$
Religion (Basegroup=Nonmuslim)				
Average (Median)	0.90 (1.00)	0.87 (1.00)	0.88 (1.00)	
Frequency (SD)	750 (0.30)	2093 (0.34)	2843 (0.33)	0.01 ${}^{4}$
Family structure (Basegroup=Nuclearfamily)				
Average (Median)	0.19 (0.00)	0.30 (0.00)	0.27 (0.00)	
Frequency (SD)	162 (0.40)	721 (0.46)	883 (0.44)	0.01 ${}^{4}$
Household eating practices (Basegroup=Others)				
Average (Median)	0.68 (1.00)	0.85 (1.00)	0.80 (1.00)	
Frequency (SD)	566 (0.47)	2047 (0.36)	2613 (0.40)	0.01 ${}^{4}$
Sample size	831	2417	3248	

${}^{1}$ The median is in parentheses. ${}^{2}$ SD stands for the standard deviation. ${}^{3}$ The Mann-Whitney test is applied to check a distributional difference of the variable between urban and rural areas. ${}^{4}$ The chi-square test is applied to examine whether or not the frequencies of the variables are independent of urban and rural areas.

**Table 4 nutrients-15-02126-t004:** Regression coefficients of the independent variables on the DDS in the ordinary Poisson and two-level random intercept Poisson regressions.

	Ordinary Poisson Regression			Two-Level Random Intercept Poisson Regression		
	Model 1-1	Model 1-2	Model 1-3	Model 2-1	Model 2-2	Model 2-3
Family role dummies (Basegroup=Fathers)						
Mothers	−0.02		−0.01	−0.02		−0.01
Sons	−0.01		0.03	−0.02		0.01
Daughters	−0.05 **		0.01	−0.04 *		0.003
Grandfathers	−0.13 **		−0.17 **	−0.12 *		−0.15 **
Grandmothers	−0.11 **		−0.13 **	−0.10 **		−0.12 **
Age group dummies (Basegroup=Adults)						
Children		−0.07 ***	−0.08 ***		−0.05 **	−0.06 **
Elderly		−0.04	0.07		−0.04	0.05
Sociodemographic variables						
Father education			0.007 ***			0.007 ***
Mother education			0.01 ***			0.01 ***
Household poverty (Basegroup=Nonpoor)			−0.14 ***			−0.14 ***
Area (Basegroup=Urban)			−0.11 ***			−0.11 ***
Total household earners			0.05 ***			0.05 ***
Occupation of the household head (Basegroup=Nonagriculture)			0.03			0.03
Religion (Basegroup=Nonmuslim)			−0.001			0.003
Family structure (Basegroup=Nuclearfamily)			0.01			0.01
Household eating practices (Basegroup=Others)			0.02			0.02
Observations	3248	3248	3248	3248	3248	3248
Groups: Household	-	-	-	811	811	811
Random effect (SD ${}^{1}$): Household	-	-	-	0.20 ***	0.20 ***	0.16 ***
Likelihood-Ratio/Wald $\chi^{2}$	12.51 **	14.04 ***	258.02 ***	7.98	6.58 **	172.80 ***
AIC ${}^{2}$	12,686.37	12,678.85	12,464.45	12,511.65	12,507.10	12,384.42

*** significant at the 1 percent level, ** at the 5 percent level and * at the 10 percent level. ${}^{1}$ SD stands for the standard deviation. ${}^{2}$ AIC stands for the Akaike information criterion.

## Data Availability

The data presented in this study are available on request from the corresponding author. The data are not publicly available due to privacy and ethical restrictions.

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
