# Peer review of "Intrahousehold Food Intake Inequality by Family Roles and Age Groups"

_nutrients, 2023, doi:10.3390/nu15092126_

Round 1
Reviewer 1 Report
Dear Authors,
I have read the article entitled “Intrahousehold food intake inequality by family roles and age groups” with great interests. It is especially important to know the food intake of all family members, how it varies, but also what causes it. There may be cultural, economic, but also psychological reasons, such as attitudes towards food, etc. There is little research of this kind, and therefore it is fully justified to study this issue. In this article, attention was primarily paid to the dietary diversity due to family roles and age. The article is very large in terms of volume and a bit wordy, so it could be shortened a bit by limiting some fragments. In this way, attention would be more focused on the results and their interpretation.
I have a few remarks regarding the paper with varying degrees of detail. Here they are:
1/ In the Result chapter there are discussion elements (for example lines 402-414), but also methodological elements (information about statistical analysis – for example lines 310-312 or 355-316) that should be moved to the appropriate sections. Thanks to this, this part of the paper will be reduced, and it will more clearly present the results of the study.
2/ The last paragraph in which the conclusions from the study will be presented has not been separated in the article. Conclusions are included in various places of discussion, but they are missing at the end of the article.
3/ I propose a change in the way of presenting the results, namely a more synthetic way of describing them, but also changing the titles of the tables, so that it is clear from the very beginning what is presented in these tables.
4/ In the methodology it is written that Chi squared test was used to confirm the differences between the city and the countryside, and then under table 2 there is information that it is used to confirm the differences resulting from family roles. This needs to be supplemented in the methodology.
If Chi squared test was applied to examine whether or not the frequencies of the food groups are independent of family roles / urban or rural residence the question appears: why mean values of the frequency are presented in the table? .
Some very detailed comments:
Lines 80-83. This information should be placed in Methodology section.
Line 141. “All The first author” – it should be changed.
Line 303. There is a lack of “Figure”.
Line 338. What does it mean “ Statistically and economically significant”?
Line 509. I think that should be “Food diversity” instead of “food intake”
Reviewer 2 Report
This paper “Intrahousehold food intake inequality by family roles and age groups” is an interesting study and a very important topic to discuss.
I have some methodological points and suggestions to make the paper easier to read.
Please change in the text gender by sex, and calorie by energy.
It would be valuable if the authors could supply as a supplement the theoretical calculations regarding the expenditure of a food bundle (line 179, “At first, the expenditure of a food bundle is computed, which includes rice, wheat, pulses, milk, oil, meat, fish, potato, other vegetables, sugar, egg, spices, fruits and others. In the next, nonfood expenditures of household is calculated and added to the food expenditure to get the total expenditure of a household and divided by the household size to have per capita expenditure per month”).
The authors mentioned in line 297 “In table 3, some of the variables, such as total household earners and religion, are statistically significant even though their mean difference is very small.”- please add the directions of the differences.
In Table 2, since no frequencies/prevalences of eating each of the food groups (more interesting than the “average} that was presented), authors could reorganize this table or add the information (frequencies/prevalences of eating each of the food groups) in a supplementary table.
Some words are missing: lines 303, 304, 313,
The results section is too long, and difficult to read and should be reorganized in a more concise way.
The sections in lines 327 - 336 and 373|384 are about methods and should be moved accordingly. Avoid phrases in the results discussing the results (line 355, “The results indicate that poor people have lower dietary diversity than nonpoor people, being consistent with the past literature [13].”.
Separe the sections discussion and conclusions.
Round 2
Reviewer 1 Report
Dear Authors,
Thank you for taking into account my suggestions. I accept all the changes.
However, I still have two suggestions. In my opinion the lines 511 - 518 with general information on the research should be deleted in Conclusions. Moreover, information on limitations of the study (lines 530-546) should be rather in the Discussion (at the end) than in Conclusion.
Kind regards
Reviewer
Reviewer 2 Report
The authors addressed the questions raised in the review.
